# The Cardiac Comeback-Beating Stronger: Exploring the Remarkable Resilience of the Heart in COVID-19 Recovery through Cardiac Autonomic Analysis

**DOI:** 10.3390/medicina59081397

**Published:** 2023-07-30

**Authors:** Sakshi Garg, Preeti Saini, Moattar Raza Rizvi, Fuzail Ahmad, Mohammed AlTaweel, Zia Ul Sabah, Humayoun K. Durrani, Shatha A. Almasswary, Mohamed K. Seyam, Ankita Sharma, Irshad Ahmad, Sara Al Marzoogi, Mohammad A. Shaphe, Shadab Uddin, Irshad Ahmad

**Affiliations:** 1Department of Physiotherapy, School of Allied Health Sciences, Manav Rachna International Institute and Studies (MRIIRS), Faridabad 121001, India; sakshigarg017@gmail.com (S.G.); preeti.fas@mriu.edu.in (P.S.); ankitasharma.fas@mriu.edu.in (A.S.); irshad.fahs@mriu.edu.in (I.A.); 2Department of Physiotherapy, School of Allied Health Sciences, MVN University, Palwal 121102, Haryana, India; 3School of Allied Health Sciences, Manav Rachna International Institute and Studies (MRIIRS), Faridabad 121001, India; mrrizvi.fas@mriu.edu.in; 4Respiratory Care Department, College of Applied Sciences, Almaarefa University, Diriya 13713, Riyadh, Saudi Arabia; smarzoogi@mcst.edu.sa; 5Department of Medicine, College of Medicine, King Khalid University, Abha 61413, Saudi Arabia; zsabah@kku.edu.sa (Z.U.S.); hkdurrani@kku.edu.sa (H.K.D.); 6College of Medicine, King Khalid University, Abha 61413, Saudi Arabia; shathaalmasswary@gmail.com; 7Department of Physical Therapy & Health Rehabilitation, College of Applied Medical Sciences, Majmaah University, Almajmaah 15431, Saudi Arabia; m.seyam@mu.edu.sa; 8Department of Physical Therapy, Faculty of Applied Medical Sciences, Jazan University, Jazan 45142, Saudi Arabia; mshaphe@jazanu.edu.sa (M.A.S.); ssabauddin@jazanu.edu.sa (S.U.); 9Department of Medical Rehabilitation Sciences, College of Applied Medical Sciences, King Khalid University, Abha 61413, Saudi Arabia; iabdulhamed@kku.edu.sa

**Keywords:** heart rate variability, cardiac autonomic function, COVID-19, six-minute walk test, physical performance, heart rate, blood pressure

## Abstract

*Background and Objectives*: Analyzing the cardiac autonomic function in COVID-19 patients can provide insights into the impact of the virus on the heart’s regulatory mechanisms and its recovery. The autonomic nervous system plays a crucial role in regulating the heart’s functions, such as heart rate, blood pressure, and cardiac output. This study aimed to investigate the impact of COVID-19 on heart rate variability (HRV) during a 6-min walk test (6MWT). *Materials and Methods*: The study included 74 participants, consisting of 37 individuals who had recovered from mild to moderate COVID-19 and 37 healthy controls. The study assessed heart rate variability (HRV) and blood pressure both before and after a 6-min walk test (6MWT). *Results*: The study found significant differences in a few time domains (SDNN and pNN50) and all frequency domain measures, whereas there were no significant differences in demographic characteristics or blood pressure between COVID-19-recovered individuals and healthy controls at rest. There were significant 6MWT effects on average HR, time-domain (SDNN and pNN50) measures of HRV, and all frequency domain measures of HRV. A significant group × 6MWT interaction was found for SDNN, pNN50, total power, Ln total power, LF, HF, Ln LF, Ln HF, and LF nu. *Conclusions*: Cardiac Autonomic analysis through HRV is essential to ensure the continued health and well-being of COVID-19 survivors and to minimize the potential long-term impacts of the disease on their cardiovascular system. This suggests that HRV analysis during the recovery phase following exercise could serve as a valuable tool for evaluating the physiological effects of COVID-19 and monitoring the recovery process.

## 1. Introduction

COVID-19 is a respiratory illness caused by the novel coronavirus SARS-CoV-2, which was first identified in Wuhan, China, in December 2019. It quickly spread around the globe, causing a pandemic that has affected millions of people worldwide [1]. Coronaviruses belong to a large family of viruses that can cause illness in humans and animals. Four types of coronaviruses can infect humans, including the beta-coronaviruses that cause SARS-CoV, MERS-CoV, and SARS-CoV-2. These viruses are highly contagious and can spread easily through respiratory droplets when an infected person coughs, sneezes, or talks [2]. While most people who contacted COVID-19 recovered without requiring hospitalization, some people developed severe respiratory illnesses, including pneumonia and acute respiratory distress syndrome (ARDS), which can be fatal. Despite all the efforts, the pandemic continues to affect people worldwide, highlighting the need for continued vigilance and ongoing research to better understand and control the spread of the virus [3]. 

The COVID-19 pandemic has brought unprecedented challenges for healthcare systems worldwide, with many individuals experiencing severe respiratory illness and even death as a result of infection. However, another area of concern that has emerged is the impact of COVID-19 on the heart. Studies have shown that the virus can cause a range of cardiac complications, including inflammation of the heart muscle and arrhythmias, which can lead to long-term damage and impaired heart function [4]. Despite these challenges, the heart has shown remarkable resilience in recovering from COVID-19. With appropriate medical care and lifestyle changes, many individuals have been able to regain their cardiac health and return to their normal activities. Research has also suggested that physical activity, such as aerobic exercise, may be particularly beneficial for individuals recovering from COVID-19 by improving cardiovascular health and reducing inflammation [5].

Heart rate variability (HRV) is the variation in the time intervals between successive heartbeats. HRV is a measure of cardiac autonomic function, which reflects the interplay between the sympathetic and parasympathetic branches of the autonomic nervous system (ANS) that regulate the heart rate. HRV can be measured noninvasively using electrocardiography (ECG). A high HRV indicates that the ANS is functioning well and can adapt to changes in the environment, while a low HRV indicates that the ANS is less able to adapt and may be associated with an increased risk for various health problems, such as cardiovascular disease, diabetes, and depression [6]. 

Studies have shown that COVID-19 can affect HRV in some patients. Few studies have reported that COVID-19 patients have reduced HRV, which may reflect impaired cardiac autonomic function [7]. This may be due to the direct effects of the virus on the heart or the indirect effects of systemic inflammation and stress associated with the illness. Other studies have found that HRV can be used as a predictor of disease severity in COVID-19 patients [8]. Patients with low HRV have been shown to have a higher risk of developing severe disease and requiring hospitalization. Additionally, some research suggests that HRV may be used as a tool to monitor the recovery of COVID-19 patients. HRV has been found to improve as patients recover from the illness, indicating a restoration of cardiac autonomic function [9,10].

The 6-min walk test (6MWT) is indeed considered a certified tool for assessing cardiorespiratory fitness, cardiopulmonary reserve, functional capacity, monitoring the effectiveness of treatments, and establishing the prognosis of patients with cardiopulmonary disease. The American Thoracic Society and the European Respiratory Society have published guidelines for the standardization of the 6MWT, which has helped to increase the reliability and validity of the test in clinical and research settings [11]. The test is widely used in clinical practice and research to evaluate the functional capacity of individuals with cardiopulmonary disease and monitor their response to interventions. The 6MWT is a simple, non-invasive test that is well tolerated by patients, making it a useful tool in both inpatient and outpatient settings. It provides valuable information on an individual’s physical capacity and endurance, which can be used to guide exercise and rehabilitation programs and monitor disease progression [12].

Several studies have evaluated the recovery of heart function in individuals with COVID-19 [13,14]. These studies have shown that some patients experience a decline in cardiac function during the acute phase of the illness, but most patients recover within several weeks or months. However, some patients may experience persistent cardiac dysfunction, which can lead to long-term complications. Comparing the recovery of heart function in individuals with COVID-19 to that of the non-COVID population is an important area of investigation [15]. This can help identify the unique features of COVID-19-related cardiac dysfunction and inform the development of effective treatment strategies. Overall, while the immediate changes in heart function in individuals with COVID-19 have not been extensively studied, there is growing evidence regarding the impact of the virus on the cardiovascular system and the recovery of heart function in affected individuals. Further research is needed to fully understand the extent of cardiopulmonary malfunctioning after COVID-19 and inform the development of effective treatment strategies.

This study aims to shed light on the potential impact of COVID-19 on the cardiovascular system by exploring changes in cardiac autonomic function, as measured by HRV, in individuals who have recovered from the disease. By comparing these results with those of a healthy control group, the study seeks to identify unique features of COVID-19-related cardiac complications and potential prognostic markers for long-term cardiac effects. This investigation may offer critical insights into the complex relationship between COVID-19 and the cardiovascular system and provide an opportunity to develop novel interventions for the long-term management of COVID-19-related cardiac complications.

## 2. Materials and Methods

### 2.1. Study Design

The present study had a cross-sectional comparative case-control design. A cross-sectional comparative case-control design is a research method used to compare groups of individuals with and without a particular condition or disease. This design can be useful in identifying potential risk factors for a particular condition or disease and can provide insights into potential prevention and treatment strategies.

### 2.2. Sample Size

Based on a priori sample size calculation, a power analysis for an analysis of variance that examined fixed effects, special effects, main effects, and interaction was conducted using G*Power software (ver. 3.1.9.2, Heinrich Heine-University, Düsseldorf, Germany), assuming a medium effect size of 0.25, an alpha level of 0.05, and a power of 0.80 for the F test. Based on these parameters, the total required sample size is 84 participants.

### 2.3. Participants 

Eighty-four male participants were screened based on inclusion criteria for allocation into the post-COVID-19 and healthy control groups. Six participants dropped out during the initial screening, further leaving 78 participants, which were allocated to the post-COVID group (*n* = 40) and healthy controls (*n* =38) using the random sampling method, which is a non-probabilistic sampling technique to select the sample in the same geographical area to obtain homogeneity (Figure 1). Further, following the 6MWT, three participants failed in the post-COVID group, leaving 37 participants, and one participant failed in the healthy control group, leaving 37 participants. The healthy controls were age and gender-matched healthy individuals from the community who tested negative for SAR CoV by RT-PCR of their nasopharyngeal and oropharyngeal swabs or showed no symptoms of COVID-19 infection. 

### 2.4. Inclusion Criteria 

This study included participants with COVID-19 status confirmed via nasopharyngeal and oropharyngeal swab RT-PCR test with a cycle threshold value between 25 and 35, the presence of COVID-19 symptoms, and a recovered COVID-19 infection within a 3- to 9-month period. Additionally, middle-aged participants between the ages of 31 and 55 years with a BMI ranging from 18 to 29.9 kg/m^2^ were included in the study.

### 2.5. Exclusion Criteria 

Participants with a history of heart failure, diagnosed coronary artery disease, uncontrolled hypertension, metabolic disease, autoimmune disease, percutaneous coronary intervention, musculoskeletal disorders limiting walking, and narcotics users were excluded. Patients on treatment with beta-blockers, inhaled or oral beta-mimetics, theophylline, and other drugs with potential chronotropic effects are also excluded. Patients with severe (those who required oxygen support/intensive care unit) disease having a cycle threshold value less than 25 in RT-PCR were excluded.

### 2.6. Ethical Consideration 

Ethical approval was obtained from the Ethical Committee at the Faculty of Allied Health Science, Manav Rachna International Institute of Research and Studies, following Ethical principles for Medical research involving humans (WMA Declaration of Helsinki) with Reference No.: MRIIRS/FAHS/PT/ 2022-23/C-01 dated 16 April 2021. All participants gave written informed consent to the study. They were informed about their rights as research subjects. All identifying information about the participants was kept confidential. The study purpose and procedure were explained to each participant. All doubts raised by participants were cleared regarding the procedure.

Demographic characteristics such as age, weight, height, and BMI were assessed. The participants were instructed to abstain from substances such as tobacco, alcohol, or stimulants (caffeine, theine, taurine, etc.) for 8 h. before the test and to have a regular breakfast (3 h before the test) on the day of assessment. Participants were instructed to have adequate rest—at least eight hours of uninterrupted sleep at night—and not to perform any strenuous exercise on the day before the assessment. They were instructed to wear comfortable, loose-fitting clothing. The assessment was taken in a quiet and dimly lit room at the same time (11:00 am to 1:00 pm) for both groups. The temperature of the room was kept between 22 and 24 °C.

### 2.7. Procedure

The digital ECG was recorded at baseline before the 6MWT in both post-COVID-19 and normal individuals. Following this, the participants were asked to perform the 6-min walk. Immediately after this, they were asked to sit on the armrest chair, and the digital ECG was recorded again for the next 5 min. During the recording of the ECG, participants were instructed to close their eyes, avoid any conversation, and avoid movement of their body or body parts during a sitting position [16]. 

#### 2.7.1. Heart Rate Variability 

On the day of the assessment, participants were asked to lie down supine for 10 min or more to attain complete relaxation. Their skin was prepared by using a razor to remove hair from the HR sensor, cleaning the skin, and drying it with gauze. They were asked to sit on the armrest chair. The participants were instructed to wear an HR sensor-elastic chest strap on the thorax in direct contact with the skin. RR intervals (the time elapsed between two successive R-waves of the QRS signal on the electrocardiogram) with a sampling frequency of 1000 Hz were recorded (Polar H7, Polar Electro Oy, Kempele, Finland) and connected via Bluetooth with an Android Smartphone application (Elite HRV, Asheville, NC, USA). A digital ECG was recorded for 5 min, and BP was assessed in a sitting position [17].

The series of RR intervals was used for time-domain measures of HRV. Time domain HRV indices were studied: mean of normal-to-normal RR interval (MeanRR), root mean square of successive differences between adjacent RR intervals (RMSSD), the standard deviation of normal-to-normal RR intervals (SDNN), and the proportion of NN50 (the number of pairs of successive normal-to-normal intervals that differ by more than 50 ms) divided by the total number of normal-to-normal RR intervals (PNN50). Frequency-domain indices evaluate the distribution of power spectra across different frequency bands. Frequency domain indices include the total power (TP) of the HRV signal, which represents the overall variability in the signal (0.003 to 0.4 Hz), Low-Frequency Spectral Power (LF), which is believed to reflect both sympathetic and parasympathetic activity (0.04 and 0.15 Hz), High-Frequency Spectral Power (HF), which is believed to reflect mainly parasympathetic activity (0.15 and 0.4 Hz), Low-Frequency Normalized Unit (LFnu), which is the percentage of the total power in the LF band, which is thought to reflect the sympathetic-parasympathetic balance, High-Frequency Normalized Unit (HFnu) is the percentage of the total power in the HF band, which is thought to reflect parasympathetic activity, and the Ratio of Low Frequency and High Frequency (LF/HF) is often used as an indicator of the sympathetic-parasympathetic balance, with a higher ratio reflecting greater sympathetic activity and a lower ratio reflecting greater parasympathetic activity [18].

The combination of all these variables in HRV comprehensively provides information regarding the relative contributions from the sympathetic and parasympathetic branches of the ANS to the heart. All HR recordings were visually inspected for stationarity and corrected for artifacts and ectopic beats via Kubios in-built piecewise cubic spline interpolation [19].

#### 2.7.2. Six-Min Walk Test (6MWT)

The 6-min walk test (6MWT) is a simple and commonly used test to assess a person’s exercise tolerance and functional capacity. The procedure involves recording the patient’s baseline vital signs, having them put on comfortable shoes and stand at the starting line of a 16-m hallway, and instructing them to walk as far as possible in six min while maintaining a steady pace. A cone or marker is placed at the turning point, and the patient is encouraged to walk in a straight line and avoid running or jogging. At the end of six min, the patient stops walking, and vital signs, including heart rate and blood pressure, are recorded [12].

### 2.8. Statistical Analysis

The data were examined using SPSS version 21 (IBM Corporation, Armonk, New York, United States). The Shapiro-Wilk test was used to assess the normality of the distribution of outcome measures. RMSSD, PNN50, TP, LF, HF, and LF/HF were found to be non-normally distributed and log-transformed for further analysis. Demographical characteristics and outcome measures before 6MWT were compared using an independent *t*-test for between-group differences, i.e., COVID-19 and healthy groups. A mixed 2 (before and after 6MWT) × 2 (COVID-19 and healthy group) analysis of variance (ANOVA) was used for all outcome variables to determine the main effect (6MWT effect and group effect) and 6MWT × group interaction. A *p* ≤ 0.05 was considered significant, and the confidence interval was set at 95%.

## 3. Results

The calculated sample size for the study was 84 participants, while we only evaluated 74 participants in total (37 in each group), resulting in a 12% loss of sample size. There were a few reasons for the loss of sample size in our study. Some participants dropped out for not meeting the inclusion criteria, and some refused to participate due to personal reasons, while others were unable to complete the study due to physical limitations during the 6MWT. Details of the same are presented in Figure 1. Additionally, recruitment was challenging during the COVID-19 pandemic, and we faced some difficulties in recruiting the full-calculated sample size.

Demographic characteristics were found to be non-significant between the COVID-19 and healthy subjects (Table 1). In the time domain of HRV, SDNN and pNN50 values showed significant differences between COVID-19 and healthy controls, whereas there was no significant difference in RMSSD. All frequency domain indices were significant between the two groups at rest (before 6MWT) (Table 1). These alterations may indicate changes in autonomic modulation and cardiac regulatory mechanisms in COVID-19 individuals. In addition, the two groups found systolic and diastolic blood pressure to be non-significant.

Average HR showed a statistically significant 6MWT effect (ɳp2 = 0.80, *p* < 0.001), group effect (ɳp2 = 0.35, *p* < 0.001), and Group × 6MWT interaction (ɳp2 = 0.07, *p* = 0.02). There was a statistically significant 6MWT effect for RMSSD (ɳp2 = 0.52, *p* < 0.001), ln RMSSD (ɳp2 = 0.30, *p* < 0.001), SDNN (ɳp2 = 0.56, *p* < 0.001), pNN50 (ɳp2 = 0.08, *p* = 0.01), and Mean RR (ɳp2 = 0.82, *p* < 0.001) whereas group effect was found to be non-significant for all time-domain measures of HRV except SDNN (ɳp2 = 0.11, *p* < 0.001) and pNN50 (ɳp2 = 0.13, *p* < 0.001). The group × 6MWT interaction of time-domain measures of HRV was found to be significant for only SDNN (ɳp2 = 0.31, *p* < 0.001) and pNN50 (ɳp2 = 0.06, *p* = 0.04) (Table 2).

There was a significant 6MWT effect for total power (ɳp2 = 0.22, *p* < 0.001), Ln total power (ɳp2 = 0.15, *p* < 0.001), LF (ɳp2 = 0.17, *p* < 0.001), HF (ɳp2 = 0.24, *p* < 0.001), VLF (ɳp2 = 0.20, *p* < 0.001), Ln LF (ɳp2 = 0.27, *p* < 0.001), Ln HF (ɳp2 = 0.40, *p* < 0.001), LF nu (ɳp2 = 0.22, *p* < 0.001), HF nu (ɳp2 = 0.55, *p* < 0.001), and LF/HF (ɳp2 = 0.13, *p* < 0.001), whereas group effect was found to be significant for all frequency domain measures of HRV (Figure 2). Group×6MWT interaction of frequency domain measures of HRV was found to be significant for total power (ɳp2 = 0.16, *p* < 0.001), Ln total power (ɳp2 = 0.07, *p* < 0.001), LF (ɳp2 = 0.14, *p* < 0.001), HF (ɳp2 = 0.15, *p* < 0.001), Ln LF (ɳp2 = 0.07, *p* = 0.018), Ln LF (ɳp2 = 0.12, *p* < 0.001), Ln HF (ɳp2 = 0.12, *p* < 0.001), LF nu (ɳp2 = 0.06, *p* = 0.04), and HF nu (ɳp2 = 0.07, *p* = 0.021) whereas no significant Group × 6MWT interaction was found for VLF, HFnu and LF/HF (Table 2).

Systolic BP was found to be non-significant in the 6MWT effect (ɳp2 = 0.02, *p* = 0.28), group effect (ɳp2 = 0.00, *p* = 0.87), and group × 6MWT interaction (ɳp2 = 0.00 *p* = 0.75). Diastolic BP was found to be non-significant in the 6MWT effect (ɳp2 = 0.05, *p* = 0.05, close to significant), group effect, and group × 6MWT interaction (Table 2).

This study underscores the potential of HRV analysis during exercise as a valuable tool for comprehensively assessing the physiological effects of COVID-19 and closely monitoring the recovery process. “Further research is warranted to explore the long-term implications of these findings and to investigate the potential of HRV assessment during exercise as a prognostic tool for individuals affected by COVID-19.

## 4. Discussion

The goal of the study was to compare the HRV in post-COVID-19 individuals and normal individuals after stress exercise (6MWT) to gather knowledge regarding whether there is any difference between HRV parameters (time and frequency domain) in post-COVID-19 individuals and normal individuals. HRV is a measure of the variation in time intervals between consecutive heartbeats and provides valuable insights into the functioning of the autonomic nervous system, which regulates various bodily processes [5]. This study was an attempt to determine the heart functioning between two stated populations in order to access the cardiovascular manifestation caused by a coronavirus. The result obtained from the analysis showed time (pre-post effect) × group (post-COVID and normal individuals) interaction, revealing some significance in certain parameters discussed ahead. In individuals affected by COVID-19, the analysis of heart rate variability (HRV) has shown some distinct changes compared to healthy individuals. Reduced SDNN suggests decreased overall HRV, while increased PNN50 indicates heightened parasympathetic activity. The stability of RMSSD suggests consistent short-term HRV. Higher values in the frequency-domain parameters indicate increased sympathetic activity. These changes reflect the impact of COVID-19 on the autonomic nervous system, likely resulting from systemic inflammation and physiological stress [7,8,9].

The study compared the effects of the 6-min walk test (6MWT) on a COVID-19 group and a healthy group. The results showed that there were minimal changes in systolic and diastolic blood pressure following the 6MWT in both groups. The mechanism of action behind these findings is unclear based on the provided information. However, it is possible that the 6MWT, being a moderate-intensity exercise, may not have exerted significant acute effects on blood pressure in this context. This may have been due to factors such as differences in age ranges or instrument calibration, which may have influenced the blood pressure. In one study, women with post-acute COVID-19 syndrome had impaired chronotropic responses during and after the 6MWT, which indicates a reduced ability of the heart to increase its rate in response to physical activity [20]. In another study that assessed the impact of mild-to-moderate COVID-19 on the cardiorespiratory fitness of young and middle-aged individuals, it was reported that after recovering from COVID-19, participants had a lower VO2max and maximal work rate, which could be related to changes in blood pressure. However, the study did not report any direct measurements of blood pressure [21].

During our examination of the heart rate (HR) data, we noticed a sudden increase in the HR of the healthy group. This could be indicative of their higher cardiac strength, which could be reflected in the greater amplitude of the QRS complex. Younger individuals, who tend to exhibit greater QRS complex amplitude, were included in our study to maintain age matching with the COVID-19 group. However, it is important to acknowledge that due to the limited number of younger participants, the generalizability of our findings regarding cardiac strength and its influence on HR responses may be somewhat restricted. Future studies with a larger sample size, specifically targeting younger individuals, would enable a more comprehensive assessment of the interplay between cardiac strength, QRS complex amplitude, and HR responses. Nevertheless, the observed increase in HR in the healthy group suggests an augmented cardiovascular response to immediate exercise, potentially driven by their robust cardiac strength and efficient pumping of blood. On the other hand, individuals who had previously contracted COVID-19 experienced cellular-level stress due to the virus’s pathological manifestation and had their HR increase proportionately more compared to the healthy group. This indicates a significant time effect and time-group interaction, as evidenced by our analysis. One study compared measurements of sympathetic nervous system activity and cardiovascular function in a group of 20 young adults who had recovered from COVID-19 with a control group of 20 healthy individuals. The results suggest that COVID-19 may lead to long-lasting effects on sympathetic nervous system activity, as evidenced by increased resting heart rate and reduced HRV in the COVID-19 group compared to the control group [22]. Another study conducted on patients admitted to the ICU with ARDS caused by COVID-19 suggests that heart rate and BP may provide useful indicators for the detection of COVID-19 in patients with ARDS. The study highlights the importance of monitoring vital signs in patients with ARDS and provides insights into potential methods for the early detection of COVID-19 [23].

Regarding RMSSD, it has been suggested that this parameter reflects parasympathetic nervous system (PNS) activity, which is responsible for the body’s rest-and-digest response. Higher RMSSD values have been associated with lower stress levels in several studies [24,25]. In the context of administering a stress exercise, it was observed that there was a significant time effect with a reduction in RMSSD values from pre- to post-exercise, thereby indicating an increase in stress levels over time. No significant interaction was found in the time effect (6MWT), indicating that the reduction in RMSSD occurred both in the normal and COVID-19 groups. This phenomenon may be attributed to the presence of cellular-level stress that could result in greater demand for cellular or molecular-level activity during exercise, which in turn may disrupt the sympathovagal tone [26]. Conversely, in the normal group, there was no discernible impact on sympathovagal tone, which resulted in a rapid decline in RMSSD values. One study examined the association between HRV and COVID-19 severity. The study found that RMSSD was significantly reduced in COVID-19 patients with severe disease compared to those with mild disease. The study suggests that HRV, including RMSSD, may be a useful tool for predicting disease severity and outcomes in COVID-19 patients [27]. Another study investigated the association between HRV and inflammation in COVID-19 patients. The study found that RMSSD was significantly correlated with inflammatory markers such as C-reactive protein and interleukin-6. The study suggests that HRV, including RMSSD, may be a potential non-invasive marker for monitoring inflammation and predicting clinical outcomes in COVID-19 patients [8].

Regarding SDNN, this parameter reflects both PNS and sympathetic nervous system activity, and higher values are thought to reflect better stress tolerance [28]. In our observational study, we found a significant difference in the time effect, indicating that stress tolerance levels decreased after performing a stress exercise. However, if an individual undergoes training through this workout, their stress tolerance levels may increase. The Mean RR interval is the duration of time between two successive R-waves, and it is inversely proportional to the HR. After any workout, the HR typically increases, leading to a decrease in the RR interval because of their inverse relationship. In our study, we observed a decrease in baseline and post-RR interval values, but we did not observe any interaction effect. Since we incorporated the 6MWT and compared the data before and after the exercise, it was expected that the RR interval value would decrease as the HR increased. We observed this effect in both the post-COVID-19 group and the normal group in our study. Previous studies reported that SDNN was significantly lower in COVID-19 patients compared to healthy controls, suggesting reduced SDNN may be associated with autonomic dysfunction in COVID-19 patients [29]. Another study investigated the association between SDNN and disease severity in COVID-19 patients. The study found that SDNN was significantly lower in COVID-19 patients with severe disease compared to those with mild disease [30].

Total power (TP) reflects the body’s state of relaxation or stress, and our study revealed both a significant Time effect and Time x group interaction. We know that the value of TP increases when the body relaxes and decreases when it is under stress, and thus the decline in pre-post values after the incorporation of 6MWT was apparent. However, cytokine elements were already present in the bodies of the post-COVID-19 group, which may have contributed to the slow decline in TP values compared to the normal group. Similar results supporting this finding were conducted in a group of 12 COVID-19 patients and found that decreased HRV was associated with increased levels of inflammation, as measured by C-reactive protein (CRP) and interleukin-6 (IL-6) levels. The study suggests that HRV may be a useful tool as a predictive marker for acute inflammatory response in COVID-19 patients [9,31].

The ratio of LF to HF power (LF/HF Ratio), normalized units of low frequency (LFnu) and high frequency (HFnu) reflect the parasympathetic and sympathetic relationships in the body [32]. When the body is under stress, the sympathetic system is activated, while the parasympathetic system is activated when the body is relaxed. After performing the exercise, it was evident that the sympathetic system was activated, leading to an increase in the value of the LF/HF ratio from the pre-post. In the Time*Group interaction, the increment of the LF/HF ratio in the post-COVID-19 group was less, likely because the value of HF was lower, increasing the value of LF. In the healthy group, it was found that the value of LF was lower due to the absence of the cytokine elements that are present in the post-COVID group. Similar results were reported in one of the studies, showing reductions in both high-frequency (HF) and low-frequency (LF) HRV parameters, suggesting impairment of both parasympathetic and sympathetic autonomic functions. These findings suggest that COVID-19 can affect the autonomic nervous system and may lead to long-term cardiac complications, highlighting the need for further research in this area [26]. This study underscores the potential of HRV analysis during exercise as a valuable tool for comprehensively assessing the physiological effects of COVID-19 and closely monitoring the recovery process. Further research is warranted to explore the long-term implications of these findings and to investigate the potential of HRV assessment during exercise as a prognostic tool for individuals affected by COVID-19.

The study had several limitations that needed to be considered. Firstly, the sample size was relatively small and consisted of only male participants. Although there was a large turnout, many individuals refused to participate, and therefore, the results may require confirmation in a larger population of each gender. Secondly, the study was performed several months after the COVID-19 infection, which may have been a limitation. Future studies should consider collecting data every month to assess the variability and incorporate some heart-related exercises for patients. Another limitation of this study was that the time and frequency domain of HRV measurement were lower for healthy controls. Our study included middle-aged health control participants with a mean age of 40 years who were free from known health conditions. HRV measurements were taken in a sitting position, which may result in lower overall HRV and increased sympathetic activity. Our study, which included only male participants, is consistent with literature indicating that males generally have lower HRV compared to females, possibly due to hormonal differences. Additionally, the participants faced challenges during the COVID-19 lockdown, which could influence HRV. This may be because of the psychological reasons during COVID-19. In future studies, more detailed assessments of psychological stress are needed to better understand the relationship between COVID-19 and HRV in healthy individuals. Furthermore, HRV was measured in a short time, and therefore, the study did not show the effect of different age groups and times of COVID infection on HRV. Finally, the use of a 16 m hallway may have led to an increase in turning points, which could have altered the walking speed of the participants. Although the study attempted to create a laboratory environment for HRV measurement, there are chances that the variability in the data could be due to the lack of a proper environment. In the study, it was discovered that the LFnu and HFnu parameters demonstrated comparable results to the TP parameter, as all three parameters are connected.

## 5. Conclusions

Individuals who have previously contracted COVID-19 may have persistent cellular-level stress, which could affect their sympathovagal response to exercise and influence their HRV parameters. The healthy group showed a greater increase in heart rate and decrease in stress levels (as measured by RMSSD) than the post-COVID-19 group. Both groups showed a decrease in stress tolerance levels (as measured by SDNN) and an increase in sympathetic activation (as measured by the LF/HF ratio) after the exercise. The post-COVID-19 group showed a slower decline in TP values compared to the normal group. Given the severity of this pandemic and the widespread impact it has had on humanity, future studies should consider collecting data every month to assess the variability and incorporate some heart-related exercises.

## Figures and Tables

**Figure 1 medicina-59-01397-f001:**
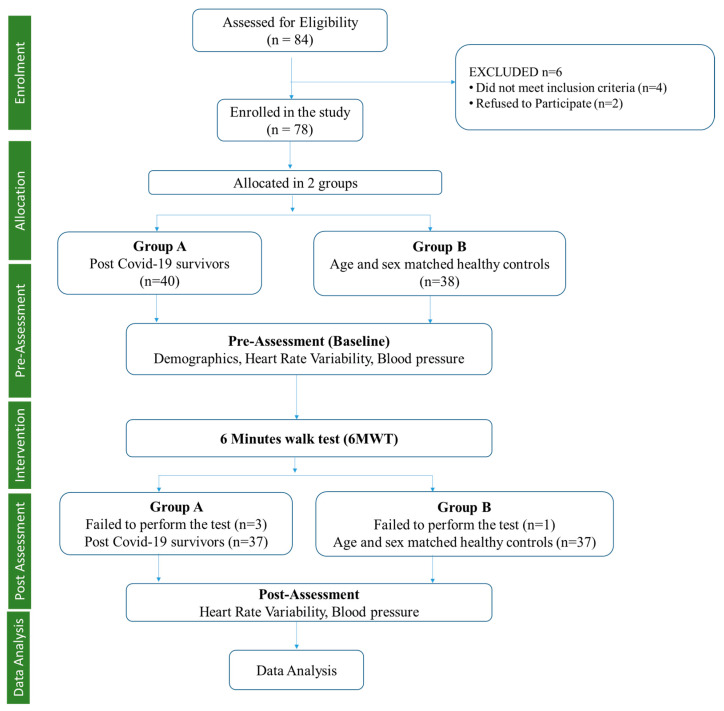
Study design.

**Figure 2 medicina-59-01397-f002:**
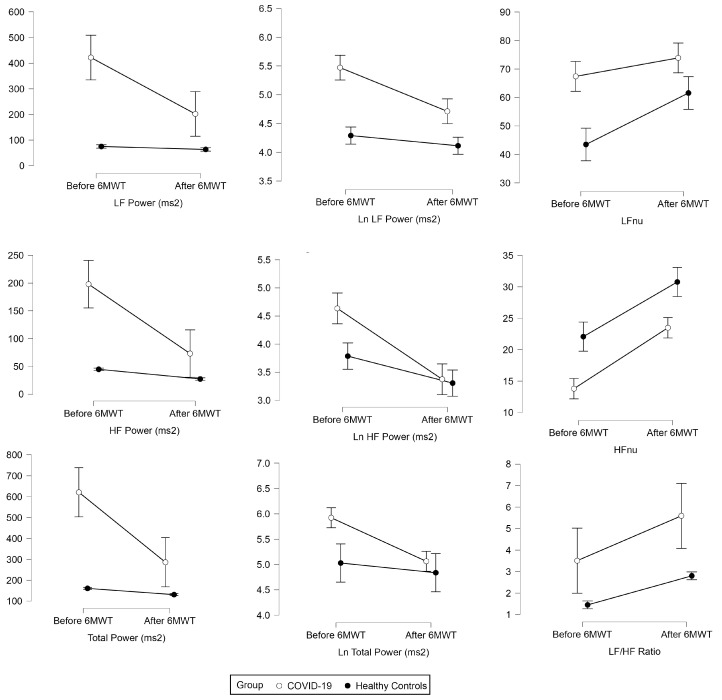
Measures of frequency domain indices of heart rate variability before and after a six-minute walk test for COVID-19 and a healthy group. LF—low frequency spectral power; HF—high frequency spectral power; TP—total power; LF/HF—ratio of low frequency and high frequency spectral power; nu—normalized units; Ln—natural logarithm.

**Table 1 medicina-59-01397-t001:** Demographic characteristics and outcome measures before the six-minute walk test of the COVID-19 group and the healthy group.

	COVID-19 Group (*n* = 37)	Healthy Group (*n* = 37)	*t* Value	*p*-Value
	Demographics			
Age (year)	41.35 ± 15.12	38.97 ± 14.51	0.69	0.49
Height (cm)	163.83 ± 8.47	163.98 ± 8.43	−0.08	0.94
Weight (kg)	69.03 ± 10.54	70 ± 9.52	−0.42	0.68
BMI (kg/m^2^)	25.66 ± 3.14	25.95 ± 2.54	−0.45	0.65
Gender (Male)	37	37	-	-
Time since COVID infection (month)	6.24 ± 1.38	-	-	-
Average HR (bpm)	90.51 ± 10.59	80.11 ± 6.29	5.14	*p* < 0.001
	Time domain measures of HRV			
MeanRR (ms)	696.4 ± 89.34	672.04 ± 75.91	1.26	0.21
RMSSD (ms)	22.61 ± 12.48	20.31 ± 7.8	0.95	0.34
Ln RMSSD (ms)	2.92 ± 0.67	2.96 ± 0.64	−0.29	0.77
SDNN (ms)	39.12 ± 16.73	58.91 ± 19.11	−4.74	*p* < 0.001
pNN50 (%)	6.1 ± 8.82	1.05 ± 0.89	3.47	*p* < 0.001
	Frequency domain measures of HRV			
Total Power (ms^2^)	621.35 ± 622.46	160.8 ± 66.49	4.48	*p* < 0.001
Ln Total Power (ms^2^)	5.92 ± 1.15	5.02 ± 1.2	3.26	*p* < 0.001
LF (ms^2^)	422.18 ± 461.24	75.26 ± 16.57	4.57	*p* < 0.001
HF (ms^2^)	198.05 ± 195.15	44.73 ± 9.84	4.77	*p* < 0.001
VLF (ms^2^)	55.57 ± 12.57	36.68 ± 7.93	7.73	*p* < 0.001
Ln LF (ms^2^)	5.47 ± 1.27	4.29 ± 0.41	5.36	*p* < 0.001
Ln HF (^ms2^)	4.63 ± 1.35	3.78 ± 0.35	3.69	*p* < 0.001
LF nu	67.41 ± 18.86	43.49 ± 18.63	5.49	*p* < 0.001
HF nu	13.79 ± 15.36	22.08 ± 6.3	−3.04	*p* < 0.001
LF/HF	3.5 ± 3.65	1.45 ± 0.54	3.38	*p* < 0.001
	Blood Pressure			
Systolic	131.41 ± 12.64	131.49 ± 14.36	−0.03	0.98
Diastolic	86.19 ± 10.04	85.81 ± 11.23	0.15	0.88

HR—Heart rate; HRV—Heart rate variability; MeanRR—mean RR normal-to-normal interval; RMSSD—root mean square of successive differences between adjacent RR intervals; SDNN—standard deviation of normal-to-normal intervals; PNN50—proportion of NN50 (number of pairs of successive normal-to-normal intervals that differ by more than 50 ms) divided by the total number of normal-to-normal intervals); LF—low frequency spectral power; HF—high frequency spectral power; LF/HF—ratio of low frequency and high frequency spectral power; nu—normalized units; Ln—natural logarithm.

**Table 2 medicina-59-01397-t002:** Time and Frequency domain measures of HRV and blood pressure measured before and after a six-minute walk test in the COVID-19 group and the healthy group.

Outcomes	COVID-19 Group (*n* = 37)		Healthy Group (*n* = 37)		Effect of 6MWT	Group Effect	GroupX6MWT Interaction
	6MWT Pretest ScoreMean ± SD	6MWT Posttest Score Mean ± SD	6MWT Pretest Score Mean ± SD	6MWT Posttest Score Mean ± SD	*p*-Value	*p*-Value	*p*-Value
HR	90.51 ± 10.59	105.86 ± 12.68	80.11 ± 6.29	91.7 ± 4.1	<0.001 *	<0.001 *	0.02
Time domain measures of HRV							
Mean RR	696.4 ± 89.34	574.64 ± 95.94	672.04 ± 75.91	542.16 ± 73.18	<0.001 *	0.12	0.56
RMSSD	22.61 ± 12.48	14.44 ± 9.84	20.31 ± 7.8	12.14 ± 7.79	<0.001 *	0.26	1
Ln RMSSD	2.92 ± 0.67	2.43 ± 0.77	2.96 ± 0.64	2.47 ± 0.7	<0.001 *	0.77	0.96
SDNN	39.12 ± 16.73	32.35 ± 15.17	58.91 ± 19.11	31.73 ± 10.6	<0.001 *	<0.001 *	<0.001 *
PNN50 (%)	6.1 ± 8.82	2.66 ± 6.79	1.05 ± 0.89	0.74 ± 1.42	0.01	<0.001 *	0.04
Frequency domain measures of HRV							
Total Power (ms^2^)	621.35 ± 622.46	286.09 ± 322.11	160.8 ± 66.49	130.92 ± 60.88	<0.001 *	<0.001 *	<0.001 *
Ln Total Power (ms^2^)	5.92 ± 1.15	5.06 ± 1.14	5.02 ± 1.2	4.83 ± 1.15	<0.001 *	0.02	0.03
LF (ms^2^)	422.18 ± 461.24	202.15 ± 207.05	75.26 ± 16.57	63.72 ± 20.32	<0.001 *	<0.001 *	<0.001 *
HF (ms^2^)	198.05 ± 195.15	73.1 ± 129.3	44.73 ± 9.84	27.28 ± 4.26	<0.001 *	<0.001 *	<0.001 *
VLF (ms^2^)	55.57 ± 12.57	46.34 ± 16.72	36.68 ± 7.93	29.34 ± 7.99	<0.001 *	<0.001 *	0.63
Ln LF (ms^2^)	5.47 ± 1.27	4.71 ± 1.19	4.29 ± 0.41	4.11 ± 0.47	<0.001 *	<0.001 *	<0.001 *
Ln HF (ms^2^)	4.63 ± 1.35	3.37 ± 1.38	3.78 ± 0.35	3.3 ± 0.83	<0.001 *	0.04	<0.001 *
LF nu	67.41 ± 18.86	73.89 ± 19.37	43.49 ± 18.63	61.57 ± 21.66	<0.001 *	<0.001 *	0.04
HF nu	13.79 ± 15.36	23.49 ± 17	22.08 ± 6.3	30.78 ± 9.52	<0.001 *	<0.001 *	0.62
LF/HF	3.5 ± 3.65	5.59 ± 6.11	1.45 ± 0.54	2.8 ± 0.59	<0.001 *	<0.001 *	0.49
Blood Pressure							
Systolic	131.41 ± 12.64	132.32 ± 14.18	131.49 ± 14.36	133.19 ± 11.76	0.28	0.87	0.75
Diastolic	86.19 ± 10.04	84.11 ± 8.92	85.81 ± 11.23	84.51 ± 8.47	0.05	0.99	0.65

* HR—Heart rate; HRV—Heart rate variability; MeanRR—mean RR normal-to-normal interval; RMSSD—root mean square of successive differences between adjacent RR intervals; SDNN—standard deviation of normal-to-normal intervals; PNN50—proportion of NN50 (number of pairs of successive normal-to-normal intervals that differ by more than 50 ms) divided by the total number of normal-to-normal intervals; LF—low frequency spectral power; HF—high frequency spectral power; LF/HF—ratio of low frequency and high frequency spectral power; nu—normalized units; Ln—natural logarithm.

## Data Availability

The data presented in this study are available on request from the corresponding author. The data are not publicly available due to privacy restrictions.

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
