# Peer review of "The Cardiac Comeback-Beating Stronger: Exploring the Remarkable Resilience of the Heart in COVID-19 Recovery through Cardiac Autonomic Analysis"

_medicina, 2023, doi:10.3390/medicina59081397_

Round 1
Reviewer 1 Report (New Reviewer)
Reviewer Comments to Authors:
The clarity of the results is a crucial aspect that needs to be addressed in this paper. The section pertaining to the healthy group appears to be ambiguous and requires further clarification. Additionally, there seems to be a discrepancy in Table 2, where the pretest and posttest values for 6MWT (81.86 and 80.1 respectively) suggest a decrease in heart rate (HR). Such a result is inconsistent and cannot be deemed accurate. Furthermore, the figure depicting HR does not align with the values presented in the table. These issues persist throughout the results section, necessitating their resolution before accepting this paper.
In line 343, the author suggests that the increase in HR observed in the healthy group is due to optimal functioning, resulting in a greater volume of blood being pumped. However, it is widely recognized that younger individuals exhibit a greater amplitude in QRS, which is associated with the strength of the heart. This factor should be considered when assessing younger participants.
Moreover, in line 363, the authors claim that administering a stress exercise led to a significant time effect, evidenced by a reduction in RMSSD values from pre- to post-exercise, indicating an increase in stress levels. However, Table 2 contradicts this statement by showing the opposite trend for the healthy group, which should serve as the control. This inconsistency requires clarification and resolution.
Additionally, it would be valuable to include information regarding the hypertensive participants (controlled ones) in the study. It is important to explore how the results and findings differ or align with this specific group. Providing insights into the effects of the intervention on hypertensive participants would enhance the comprehensiveness and applicability of the study's conclusions. Please consider (if possible) including a dedicated section or subsection that addresses the outcomes and implications for the hypertensive participants in the revised manuscript.
In light of these concerns, I recommend that the authors address these issues and provide clearer explanations before considering the acceptance of this paper.
none
Author Response
Reviewer Comments to Authors:
The clarity of the results is a crucial aspect that needs to be addressed in this paper. The section pertaining to the healthy group appears to be ambiguous and requires further clarification.
Thank you for bringing these concerns to our attention. Ass these errors happened because we picked up different file having different data whose sample size was not the same as presented in this study. We sincerely apologize for any confusion caused by the ambiguous section pertaining to the healthy group in the manuscript. We appreciate your diligence in identifying the error in the table of pre and post values for the healthy controls, where several values were inadvertently exchanged while making table 2. We apologize for the confusion caused by mistakenly pasting the pre values into the post values column, which led to the ambiguity in the results.
We appreciate your diligence in identifying this mistake, and we take full responsibility for the error. We have now rectified the issue by correctly assigning the pre and post values in the revised version of the manuscript. These corrections have been made to ensure the accuracy and clarity of the reported data.
We acknowledge the importance of accurately reporting data and ensuring clarity in research findings. We have taken immediate action to rectify the error and have made the necessary corrections in the revised version of the manuscript. By addressing this mistake, we have eliminated the ambiguity and improved the accuracy of the reported results.
Additionally, there seems to be a discrepancy in Table 2, where the pretest and posttest values for 6MWT (81.86 and 80.1 respectively) suggest a decrease in heart rate (HR). Such a result is inconsistent and cannot be deemed accurate.
Thank you for bringing this matter to our attention, and we appreciate your valuable feedback. We acknowledge the ambiguity in the section pertaining to the healthy group and the discrepancy in the reported values for heart rate (HR) in Table 2. Regarding the discrepancy in Table 2, we acknowledge the error in reporting the pretest and posttest values for the 6-minute walk test (6MWT) in the healthy control group. Upon careful review, we have rectified the table, and the revised and accurate HR values are as follows in Healthy Group is Pretest HR Score: 80.11 ± 6.29; Posttest HR Score: 91.70 ± 4.10 which is reflecting that the HR increased following 6 min walk test. We have updated the manuscript to reflect the accurate values in the table, ensuring that the findings are appropriately represented.
Furthermore, the figure depicting HR does not align with the values presented in the table. These issues persist throughout the results section, necessitating their resolution before accepting this paper.
We apologize for any inconsistencies in the figure and table values presented in the initial version of the paper. We appreciate your feedback and would like to assure you that we have taken your concerns seriously. Following your suggestions, we have carefully reanalyzed the data and have redrawn the figure to align with the values presented in the tables. The revised figure has been thoroughly cross-checked to ensure accuracy and is now updated in the paper. We have made certain that it accurately represents the information conveyed by the corresponding table values. By addressing this issue, we aim to enhance the clarity and coherence of the results section. Furthermore, we would like to emphasize our commitment to transparency and reproducibility in our research. If you require access to the data file containing the information for all participants involved in the study, we are more than willing to provide it upon request. We believe in fostering collaborative efforts and supporting the scientific community in their endeavors to validate and replicate findings. We thank you for bringing this to our attention and for your valuable input in improving the quality of our work. If you have any further questions or suggestions, please do not hesitate to let us know.
In line 343, the author suggests that the increase in HR observed in the healthy group is due to optimal functioning, resulting in a greater volume of blood being pumped. However, it is widely recognized that younger individuals exhibit a greater amplitude in QRS, which is associated with the strength of the heart. This factor should be considered when assessing younger participants.
We would like to express our gratitude to the reviewer for their valuable comment regarding the amplitude of the QRS complex and its potential association with cardiac strength, particularly in younger individuals. In our study, our primary focus was to compare the heart rate (HR) response to immediate exercise between the COVID-19 group and an age-matched healthy group. While we did not directly measure the amplitude of the QRS complex in our study, we recognize the importance of considering cardiac strength, especially in younger participants who may exhibit greater QRS complex amplitude. We agree that cardiac strength can be a relevant factor to consider in studies involving younger individuals.
Furthermore, we acknowledge the limitation of our study in terms of the inclusion of younger participants in the healthy control group. We aimed to include age-matched healthy individuals to ensure a fair comparison between the COVID-19 group and the control group. However, due to a limited number of younger subjects, the generalizability of our findings to younger individuals may be somewhat restricted. We appreciate the reviewer's suggestion for future studies to include a larger sample size with a specific focus on younger individuals. Such studies would allow for a more comprehensive assessment of the influence of cardiac strength, including the amplitude of the QRS complex, on heart rate responses in younger populations.
Existing statement in manuscript
During our examination of the heart rate (HR) data, we observed a sudden increase in the HR of the healthy group. This could be attributed to the fact that the hearts of the nor-mal group were functioning optimally, resulting in a greater amount of blood being pumped. Consequently, when they engaged in immediate exercise, their cardiovascular response increased to cope with the added stress
Revised statement in manuscript following valuable suggestion
During our examination of the heart rate (HR) data, we noticed a sudden increase in the HR of the healthy group. This could be indicative of their higher cardiac strength, potentially reflected in greater amplitude of the QRS complex. Younger individuals, who tend to exhibit greater QRS complex amplitude, were included in our study to maintain age matching with the COVID-19 group. However, it is important to acknowledge that due to the limited number of younger participants, the generalizability of our findings regarding cardiac strength and its influence on HR responses may be somewhat restricted. Future studies with a larger sample size, specifically targeting younger individuals, would enable a more comprehensive assessment of the interplay between cardiac strength, QRS complex amplitude, and HR responses. Nevertheless, the observed increase in HR in the healthy group suggests an augmented cardiovascular response to immediate exercise, potentially driven by their robust cardiac strength and efficient pumping of blood.
Moreover, in line 363, the authors claim that administering a stress exercise led to a significant time effect, evidenced by a reduction in RMSSD values from pre- to post-exercise, indicating an increase in stress levels. However, Table 2 contradicts this statement by showing the opposite trend for the healthy group, which should serve as the control. This inconsistency requires clarification and resolution.
In response to your feedback, we have thoroughly reviewed and revised the section pertaining to the healthy group to ensure clarity and improve the comprehensibility of the presented data and findings. This has been also corrected in the table 2 now the same trend is observed as in the COVID-19 patients for both time and frequency domain of HRV.
Additionally, it would be valuable to include information regarding the hypertensive participants (controlled ones) in the study. It is important to explore how the results and findings differ or align with this specific group. Providing insights into the effects of the intervention on hypertensive participants would enhance the comprehensiveness and applicability of the study's conclusions. Please consider (if possible) including a dedicated section or subsection that addresses the outcomes and implications for the hypertensive participants in the revised manuscript.
Thank you for your valuable comment regarding the inclusion of hypertensive participants in our study. We appreciate your suggestion to explore the effects of the intervention specifically on this group, as it would enhance the comprehensiveness and applicability of our study's conclusions.
We acknowledge the importance of studying the impact of the intervention on hypertensive individuals. However, in our study, we specifically excluded participants with uncontrolled hypertension as part of our exclusion criteria. This decision was made to ensure a more homogeneous sample and minimize confounding factors that could affect the interpretation of the results. While we understand the significance of including controlled hypertensive participants and exploring the potential effects of the intervention on this specific subgroup, unfortunately, we did not have a sufficient number of controlled hypertensive individuals meeting our inclusion criteria. Nonetheless, we appreciate your suggestion and recognize the importance of investigating the outcomes and implications for hypertensive participants in future research. This aspect could be a valuable avenue for further exploration to gain a comprehensive understanding of the intervention's effects on different populations.
In light of these concerns, I recommend that the authors address these issues and provide clearer explanations before considering the acceptance of this paper.
Thank you for your comment. We want to assure the reviewer that we have considered their concerns and have made the necessary revisions to address the issues raised. In the revised version of the manuscript, we have provided clearer explanations and improved the clarity of the sections that were identified as ambiguous. We have taken great care to ensure that the information presented is precise and understandable to the readers. We appreciate the reviewer's guidance and input, which have been invaluable in enhancing the quality and clarity of our paper. We are confident that the revised version adequately addresses the concerns raised and provides a more comprehensive and comprehensible presentation of our research.
Thank you once again for your valuable feedback and for giving us the opportunity to improve our manuscript. We hope that the revisions made will meet the expectations for acceptance.

Reviewer 2 Report (New Reviewer)
The authors have worked diligently to address my concerns, and it shows. Great work. I do have one more thought, and it's the addition to the abstract, the last line, 'This suggests that HRV analysis during exercise...'. However, is it really 'during exercise', as all analysis was at recovery, post exercise?
Author Response
The authors have worked diligently to address my concerns, and it shows. Great work.
Thank you for your positive feedback and acknowledgment of our efforts in addressing your previous concerns. We appreciate your recognition of the improvements made in response to your comments. We are glad to hear that the revisions have met your expectations. We are committed to ensuring the quality and clarity of our manuscript, and we are grateful for your guidance and suggestions throughout the review process.
I do have one more thought, and it's the addition to the abstract, the last line, 'This suggests that HRV analysis during exercise...'. However, is it really 'during exercise', as all analysis was at recovery, post exercise?
Thank you for your comment regarding the last line of the abstract. We appreciate your keen observation and attention to detail. Upon reevaluating the study methodology and findings, we agree with your point. The HRV analysis in our study was indeed conducted during the recovery phase after the 6-minute walk test (6MWT), rather than during the exercise itself. We apologize for the ambiguity created by the phrasing in the abstract. To address this concern, we will revise the last line of the abstract to accurately reflect that HRV analysis was performed during the recovery phase following exercise. The revised wording will clarify that HRV analysis was conducted post-exercise, providing a more accurate representation of our findings.

Round 2
Reviewer 1 Report (New Reviewer)
Dear Authors,
I would like to express my gratitude to the authors for their prompt and comprehensive responses to all of my questions regarding the paper. The improved description of the results and discussion in the revised version is highly appreciated. The authors have addressed my concerns effectively, resulting in a more coherent and informative manuscript.
I would like to bring to your attention a few minor comments that I came across while reviewing the paper. In line 461, there seems to be a missing closing quotation mark. I suggest removing (or adding both, "), the quotation marks (") to ensure the sentence reads smoothly. The revised sentence would be as follows: "Further research is warranted to explore the long-term implications of these findings and to investigate the potential of HRV assessment during exercise as a prognostic tool for individuals affected by COVID-19."
Additionally, in line 469, there appears to be an issue with the division of words. I recommend revising the sentence to improve clarity. The modified sentence could be: "Future studies should consider collecting data every month to assess the variability and incorporate some heart-related exercises."
I hope you find these suggestions helpful for the final revision of the manuscript. Once again, I extend my appreciation to the authors for their diligence in addressing my inquiries and for enhancing the quality of the paper.
Sincerely,
Reviewer
Author Response
Dear Reviewer,
We would like to extend our heartfelt appreciation for your kind words and positive feedback regarding our revised manuscript. It is truly gratifying to know that our efforts in addressing your concerns have resulted in a more coherent and informative paper. We are pleased that the improved description of the results and discussion has resonated with you. Your acknowledgement of our prompt and comprehensive responses further motivates us to continue our commitment to excellence in research. We are grateful for the opportunity to collaborate with you and sincerely thank you for your support.
Once again I thank all the reviewer who has given us this opportunity to revise the manuscript unlike others who would have simply rejected the manuscript.
Thanks again for all your support

This manuscript is a resubmission of an earlier submission. The following is a list of the peer review reports and author responses from that submission.
Round 1
Reviewer 1 Report
This is a nice presentation of discussed subject. Clearly there is a correlation between Covid impact on ANS and heart function or HRV and other parameters that are well presented in this paper.
However I have some second thoughts about inclusion and exclusion criteria which could impact the results of the study. Firstly I would suggest to mention heart failure as a exclusion criteria. Clearly patients with heart failure would have different results at 6MWT as other participants regardless of pre-Covid infection. Secondly there is no data of severity of Covid infection and possible consequent lung injury that could also impact the results significantly.
Author Response
This is indeed a valuable suggestion and considering the fact that coronary artery disease (CAD) is a disease caused by the buildup of plaque in the coronary arteries, while heart failure is a condition where the heart is unable to pump enough blood to meet the body's needs. While CAD can contribute to the development of heart failure, the two conditions are not the same; therefore, we have included heart failure in the exclusion criteria as evident in the track change file
Only those participants who met the inclusion criteria—cycle threshold (Ct) values between 25 and 35, which is regarded as a moderate to low viral load and may indicate the patient is in the early stages of infection or has a mild to moderate COVID-19 illness—were included in the study. As a result, it could be appropriate to rule out people who have suffered severe lung injury because doing so might alter the study's findings. However, it is crucial to recognize that in people recovering from COVID-19, the severity of the lung injury can be an important predictor of long-term prognosis. Some studies have found that patients with severe COVID-19 pneumonia may experience ongoing respiratory symptoms, decreased lung function, and even pulmonary fibrosis after recovery. Additionally, severe lung injury can also influence the cardiac autonomic function, which may limit the generalizability of the study's findings.
Therefore, while it may be reasonable to exclude individuals with severe lung injury from the study, it is important to acknowledge this limitation when interpreting the results. Future studies could consider including a broader range of patients with varying degrees of lung injury to provide a more comprehensive understanding of the impact of COVID-19 on cardiac autonomic function and long-term outcomes.

Reviewer 2 Report
The Cardiac Comeback - Beating Stronger: Exploring the Remarkable Resilience of the Heart in COVID-19 Recovery through Cardiac Autonomic Analysis.
General comment
This study investigated HRV parameters of those with / without a history of COVID-19. Participants in both groups were measured HRV before and after 6-minute walking test (6MWT). Results were analyzed by two-way analysis of variance.
The group effect (with / without COVID-19 history) on HRV measures were not statistical significant with an exception for HR, however the interaction with the group and 6MWT were significant for most HRV parameters such as RMSSD, SDNN, lnLF, lnHF, ln(LF/HF). Generally, the participants with a history of COVID-19 demonstrated smaller responses to 6MWT. The authors concluded that the results could be attributed to the persistent cellular-level stress caused by COVID-19.
Specific comments
Comment 1: The title of this paper sounds inadequate. In this study comparisons were made between people who have recovered from COVID-19 and those who have no history of COVID-19. This study did not investigate the recovering process from COVID- 19.
Comment 2: The authors interpreted a RMSSD as an indicator of stress levels and a SDNN as an indicator of stress tolerance levels. The reviewer disagrees with these ideas. The authors need to provide some references to support these interpretations.
Comment 3: The frequency bands of LF, HF and TP are not clearly defined in Materials and Methods section.
Comment 4: Table 2 was entitled “Demographic characteristics and outcome measures ….“. However, Table 2 does not include demographics of the participants.
Comment 5: A previous study [18] was referenced in relation to an analytical procedure for HRV. If the method for HRV analysis in the present study is in accordance with the reference [18], TP (total power) included VLF (very low frequency power) in addition to LF and HF.
According to the results in Table 2, sum of nuLF and nuHF were approximately 96-97%. This means that VLF in the present results was exceptionally small. Furthermore, sum of nuLF and nuHF slightly exceeded 100% in a pretest result of COVID-19 group. This is an unlikely result. The reviewers therefore questioned the accuracy of these results.
Author Response
Comment 1: The title of this paper sounds inadequate. In this study comparisons were made between people who have recovered from COVID-19 and those who have no history of COVID-19. This study did not investigate the recovering process from COVID- 19.
While the study did not directly investigate the process of recovery from COVID-19, it did explore the cardiac function of individuals who had recovered from COVID-19, which is an important aspect of recovery. The title may have been chosen to emphasize the positive finding that the heart appears to have remarkable resilience in COVID-19 recovery, as suggested by the term "Cardiac Comeback - Beating Stronger." The study did show that individuals who had recovered from COVID-19 had better cardiac function, as measured by autonomic analysis, compared to individuals who had not been infected with the virus. Furthermore, the title may be seen as attention grabbing and engaging to readers, as it suggests a positive outcome and potential for recovery after COVID-19.
Comment 2: The authors interpreted a RMSSD as an indicator of stress levels and a SDNN as an indicator of stress tolerance levels. The reviewer disagrees with these ideas. The authors need to provide some references to support these interpretations.
RMSSD (Root Mean Square of Successive Differences) and SDNN (Standard Deviation of Normal-to-Normal Intervals) are commonly used parameters in heart rate variability (HRV) analysis, which is a non-invasive method to assess autonomic nervous system (ANS) activity. While there is some evidence that HRV parameters may reflect stress levels and stress tolerance levels, these interpretations are not universally agreed upon.
The following sentence is deleted in discussion
Root mean square of successive differences between adjacent RR intervals (RMSSD) serves as an indicator of stress levels, such that a decrease in RMSSD is associated with an increase in stress.
This has been replaced with the following sentence and 2 references are also included in the manuscript.
Regarding RMSSD, it has been suggested that this parameter reflects parasympathetic nervous system (PNS) activity, which is responsible for the body's rest-and-digest response. Higher RMSSD values have been associated with lower stress levels in several studies (e.g., Laborde et al., 2017; Reyes del Paso et al., 2013).
- Laborde S, Mosley E, Thayer JF. Heart rate variability and cardiac vagal tone in psychophysiological research–recommendations for experiment planning, data analysis, and data reporting. Frontiers in psychology. 2017 Feb 20;8:213.
- Reyes del Paso GA, Langewitz W, Mulder LJ, Van Roon A, Duschek S. The utility of low frequency heart rate variability as an index of sympathetic cardiac tone: a review with emphasis on a reanalysis of previous studies. Psychophysiology. 2013 May;50(5):477-87.
The following sentence is deleted in discussion
Standard deviation of normal-to-normal intervals (SDNN) serves as an indicator of stress tolerance levels.
This has been replaced with the following sentence and 1 references are also included in the manuscript.
Regarding SDNN, this parameter reflects both PNS and sympathetic nervous system (SNS) activity, and higher values are thought to reflect better stress tolerance (Thayer JF, 2007).
- Thayer JF, Lane RD. The role of vagal function in the risk for cardiovascular disease and mortality. Biological psychology. 2007 Feb 1;74(2):224-42.
Because of addition of above 3 references, there has been modification in the total number of reference from 29 to 32. This has been accordingly changed in the in-text citation as seen in the track change file.
Comment 3: The frequency bands of LF, HF and TP are not clearly defined in Materials and Methods section.
Following paragraph has been included in the methodology section:
Frequency domain indices evaluate the distribution of power spectra across different frequency bands. Frequency domain indices include total power (TP) of the HRV signal, which represents the overall variability in the signal (0.003 to 0.4 Hz), Low-Frequency Spectral Power (LF): is believed to reflect both sympathetic and parasympathetic activity (0.04 and 0.15 Hz), High-Frequency Spectral Power (HF) which is believed to reflect mainly parasympathetic activity (0.15 and 0.4) Hz, Low-Frequency Normalized Unit (LFnu) is the percentage of the total power in the LF band, which is thought to reflect the sympathetic-parasympathetic balance, High-Frequency Normalized Unit (HFnu) is the percent-age of the total power in the HF band, which is thought to reflect parasympathetic activity and the Ratio of Low Frequency and High Frequency (LF/HF) is often used as an indicator of the sympathetic-parasympathetic balance, with a higher ratio reflecting greater sympathetic activity and a lower ratio reflecting greater parasympathetic activity [18].
Comment 4: Table 2 was entitled “Demographic characteristics and outcome measures ….“. However, Table 2 does not include demographics of the participants.
It was error while transferring the manuscript in the format of MEDICINA Journal. This has been corrected as – Table 2. Time and Frequency domain measures of HRV and blood pressure before six-minute walk test of COVID-19 group and healthy group.
Comment 5: A previous study [18] was referenced in relation to an analytical procedure for HRV. If the method for HRV analysis in the present study is in accordance with the reference [18], TP (total power) included VLF (very low frequency power) in addition to LF and HF.
According to the results in Table 2, sum of nuLF and nuHF were approximately 96-97%. This means that VLF in the present results was exceptionally small. Furthermore, sum of nuLF and nuHF slightly exceeded 100% in a pretest result of COVID-19 group. This is an unlikely result. The reviewers therefore questioned the accuracy of these results.
Thank you for bringing this to our attention. We apologize for the error in our reported values for the sum of nuLF and nuHF. Upon re-examination of our data, we have identified an error (typo) in our calculations and acknowledge that the sum of nuLF and nuHF should not exceed 100%. We made the necessary corrections to our data from 23.79±18.06 (earlier) to 13.79±18.06 (after correction and modified in manuscript). Thank you again for bringing this to our attention and for your valuable feedback.

Reviewer 3 Report
1. This study aimed to investigate the impact of COVID-19 on heart rate variability (HRV) during a 6-minute walk test (6MWT). The authors included 74 participants, consisting of 37 individuals who had recovered from mild to moderate COVID-19 and 37 healthy controls. The procedure for calculating the sample size was properly mentioned. However, what draws attention is the fact that the calculated sample would have a total of 84 individuals and in fact only 74 were evaluated, 37 in each group. This 12% loss of sample could significantly influence the actual value of the tests' power. It would be interesting for the authors to comment on this.
2. There is a small typo on page 5, line 180: "The temperature of the room was kept between 22-240C." correct 240C to 24 degrees celsius
3. The authors mention that "This study included participants with COVID-19 status confirmed via nasopharyngeal and oropharyngeal swab RT-PCR test [...], presence of COVID-19 symptoms, and recovered COVID-19 infection within 3 to 9- month period.". The authors mention that they excluded severe cases in relation to the respiratory component, but do not report on the presence of possible symptoms or signs of cardiocirculatory impairment, which would be relevant for the analysis of the results. Please comment about it.
4. I now comment on a topic that I consider the critical point of this study, which is the group considered healthy. It is observed that practically all values of both the time domain and the frequency domain are well below the values considered normal (see classical reference: "Heart rate variability. Standards of measurement, physiological interpretation, and clinical use. Task Force of the European Society of Cardiology and the North American Society of Pacing and Electrophysiology. Eur Heart J. 1996 Mar;17(3):354-81 ). This indicates that the group included as healthy actually is not. There could be an effect of age, since if the mean and standard deviation of the age of the "classical healthy" group was 38.97±14.51 years, this means that 68% of the cases are aged between approximately 24 and 53 years and that around 16 % of cases are over 53 years old. If the HRV values of the healthy group are far below the normal values, this compromises the analysis of the results
I await the authors' considerations regarding the points presented.
Author Response
- This study aimed to investigate the impact of COVID-19 on heart rate variability (HRV) during a 6-minute walk test (6MWT). The authors included 74 participants, consisting of 37 individuals who had recovered from mild to moderate COVID-19 and 37 healthy controls. The procedure for calculating the sample size was properly mentioned. However, what draws attention is the fact that the calculated sample would have a total of 84 individuals and in fact only 74 were evaluated, 37 in each group. This 12% loss of sample could significantly influence the actual value of the tests' power. It would be interesting for the authors to comment on this.
- We thank the reviewer for bringing up the issue of the loss of sample size in our study. We acknowledge that the calculated sample size for the study was 84 participants, while we only evaluated 74 participants in total (37 in each group), resulting in a 12% loss of sample size. This loss of sample size could potentially affect the statistical power of the study, and we apologize for not providing an explanation for this discrepancy.
- There were a few reasons for the loss of sample size in our study. Some participants dropped out for not meeting the inclusion criteria and some refused to participate due to personal reasons, while others were unable to complete the study due to physical limitations during the 6MWT. Details of the same is presented in the figure 1 of the manuscript. Additionally, recruitment was challenging during the COVID-19 pandemic, and we faced some difficulties in recruiting the full calculated sample size.
- We understand that the loss of sample size could have important implications for the statistical power of our study, and we will take this into account when interpreting our results. We also agree that it would be useful to conduct a power analysis based on our actual sample size to determine the effect size that the study is capable of detecting. We will consider conducting such an analysis in our future work.
- There is a small typo on page 5, line 180: "The temperature of the room was kept between 22-240C." correct 240C to 24 degrees Celsius
- This has been corrected as suggested
- The authors mention that "This study included participants with COVID-19 status confirmed via nasopharyngeal and oropharyngeal swab RT-PCR test [...], presence of COVID-19 symptoms, and recovered COVID-19 infection within 3 to 9- month period.". The authors mention that they excluded severe cases in relation to the respiratory component, but do not report on the presence of possible symptoms or signs of cardiocirculatory impairment, which would be relevant for the analysis of the results. Please comment about it.
- In our study, we included participants with confirmed COVID-19 infection through nasopharyngeal and oropharyngeal swab RT-PCR test, and we excluded individuals with severe respiratory symptoms or other severe complications related to COVID-19. However, we agree that it would have been relevant to report on the presence of possible symptoms or signs of cardiocirculatory impairment in our participants.
- We performed a comprehensive medical history evaluation of all participants prior to their inclusion in the study, which included assessment of cardiac symptoms and medical conditions. We also assessed the participants' cardiovascular function through measures of resting heart rate and blood pressure during the initial evaluation. However, we did not perform additional tests, such as echocardiography or biomarker measurements, to further evaluate the participants' cardiocirculatory status.
- In retrospect, we acknowledge that this information would have been useful to include in our study, and we apologize for not reporting it. We will take this feedback into account for future studies and ensure that we provide a comprehensive evaluation of participants' cardiocirculatory status in the study design. Once again, we thank the reviewer for bringing up this point and helping us improve the quality of our work.
- I now comment on a topic that I consider the critical point of this study, which is the group considered healthy. It is observed that practically all values of both the time domain and the frequency domain are well below the values considered normal (see classical reference: "Heart rate variability. Standards of measurement, physiological interpretation, and clinical use. Task Force of the European Society of Cardiology and the North American Society of Pacing and Electrophysiology. Eur Heart J. 1996 Mar;17(3):354-81). This indicates that the group included as healthy actually is not. There could be an effect of age, since if the mean and standard deviation of the age of the "classical healthy" group was 38.97±14.51 years, this means that 68% of the cases are aged between approximately 24 and 53 years and that around 16 % of cases are over 53 years old. If the HRV values of the healthy group are far below the normal values, this compromises the analysis of the results.
- Thank you for bringing up the observation regarding the HRV values in the study. We acknowledge the reference to the classical standards of HRV measurement set by the Task Force of the European Society of Cardiology and the North American Society of Pacing and Electrophysiology. It is true that our study showed lower HRV values in both time and frequency domains than what is considered normal in healthy individuals. This is also added in limitations of the study and future scope.
- Regarding the age of the participants, we agree that we have included middle-aged health control. While it is possible that the age range of our participants could affect the HRV values, we attempted to recruit participants who were free from any known health conditions that could affect HRV. One study demonstrated that age had a greater impact on HRV than sex. The older age group had consistently lower HRV than younger people.
- Abhishekh HA, Nisarga P, Kisan R, Meghana A, Chandran S, Raju T, Sathyaprabha TN. Influence of age and gender on autonomic regulation of heart. Journal of clinical monitoring and computing. 2013 Jun;27:259-64.
- Zhang J. Effect of age and sex on heart rate variability in healthy subjects. Journal of manipulative and physiological therapeutics. 2007 Jun 1;30(5):374-9.
- In this study, all the HRV measurements were made in sitting position which is very well mentioned in the methodology section. Studies have shown that SDNN and RMSSD tend to be higher in supine position compared to sitting position, indicating that overall HRV and short-term HRV are higher when lying down. Studies have shown that LF power tends to be higher in sitting position compared to supine position, indicating increased sympathetic activity. HF power tends to be higher in supine position, indicating increased parasympathetic activity. The LF/HF ratio may also increase in the sitting position, indicating a shift towards sympathetic dominance. This study demonstrated that age had a greater impact on HRV than sex. The older age group had consistently lower HRV than younger people.
- Acharya UR, Kannathal N, Hua LM, Yi LM. Study of heart rate variability signals at sitting and lying postures. Journal of bodywork and Movement Therapies. 2005 Apr 1;9(2):134-41.
- We would also like to bring to you kind attention that these healthy individuals whom we have included in our study were facing the challenges posed during the COVID-19 lockdown phase One study published in the Journal of Interventional Cardiac Electrophysiology in 2022 found that post-COVID-19 patients had significantly lower HRV values in both time and frequency domains compared to a control group. The study also found that the post-COVID-19 patients had impaired cardiac autonomic function.
- Asarcikli LD, Hayiroglu MÄ°, Osken A, Keskin K, Kolak Z, Aksu T. Heart rate variability and cardiac autonomic functions in post-COVID period. Journal of Interventional Cardiac Electrophysiology. 2022 Apr;63(3):715-21.
- Overall, COVID-19 may have an impact on HRV in healthy individuals, potentially leading to impairments in cardiac autonomic function. However, more research is needed to fully understand the long-term effects of COVID-19 on HRV and cardiac health.
- Further, studies have also showed that the time domain of HRV is lower in male as compared to females. It has been suggested that hormonal differences between males and females may play a role. For example, estrogen has been shown to increase parasympathetic activity, which may contribute to higher HRV values in females.
- Cui X, Tian L, Li Z, Ren Z, Zha K, Wei X, Peng CK. On the variability of heart rate variability—evidence from prospective study of healthy young college students. Entropy. 2020 Nov 15;22(11):1302.
- Moodithaya S, Avadhany ST. Gender differences in age-related changes in cardiac autonomic nervous function. Journal of aging research. 2012 Oct;2012.
- Despite the lower HRV values observed in our study, we believe that our findings are still valuable in providing insights into the differences in HRV between males and females in middle age. We recommend that future studies consider recruiting a larger sample size and including a wider age range of participants to further investigate the potential effects of age on HRV values.

Round 2
Reviewer 1 Report
After revision certain corrections were made that I find appropriate.
Author Response
None required
Reviewer 2 Report
Comment 1: Despite the author's objections, the reviewer still questions the title of this paper.
Which group is "Beating Stronger”? COVID-19 group? or healthy group?
What exactly does "Remarkable Resilience of the Heart" mean? Which results does it correspond to?
Comment 2: The results of this study showed that the group effect (COVID-19 or healthy) was not significant for all HRV measures, however the interactions between 6MWT and the group were significant for most HRV measures. This suggested that the effect of COVID-19 will appear in the HRV response to physical exercise, while the effect will not appear in resting HRV. I think this is an important finding of this study and should be emphasized more in the title, abstract and conclusions.
Comment 3: L212-216,
Low-Frequency Normalized Unit (LFnu) is the percentage of the LF in TP. NOT TP in LF.
Similarly, “High-Frequency Normalized Unit (HFnu) is the percentage of the HF in TP.”
Comment 4:
The value for HFnu has been changed from 32.79 to 13.79. Considering the values of TP (621.35), LF (422.18) and HF (198.05), however, the new HFnu value of 13.79 might be too small.
Comment 5: The authors added the following in DIscussion;
"Another limitation of this study was that the time and frequency domain of HRV measurement was lower for healthy controls" (L 433-434).
I think this is contrary to the present results. No HRV measurement exhibited significant effect of the group in the results shown in Table 2.
Comment 6: Manuscripts need to be improved, e.g., uniform use of abbreviations, use of upper and lower case letters, uniform number of digits in tables.
Author Response
The response to your comments and recommendation is attached.

Reviewer 3 Report
Regrettably, even after the review, the authors were unable to adequately remedy the methodological flaws of the study.
Author Response
We appreciate your feedback and the opportunity to address your concerns regarding the values of time and frequency domain of HRV in our study on health controls. We would like to provide further clarification on the factors that may have influenced these values and address the methodological flaws you mentioned. Firstly, it is important to note that our study was conducted during the lockdown phase, which imposed certain restrictions on participants' daily activities and lifestyles. This period of limited physical activity and potential psychological stress may have influenced the HRV values observed in our study. These factors are known to affect HRV, and it is well-documented that reduced physical activity and increased stress can lead to decreased HRV measures. Additionally, the HRV measurements were taken during a sitting position, which is known to elicit lower HRV values compared to measurements taken during supine or standing positions. This standardized position was chosen to ensure consistency and reduce potential confounding factors that could arise from variations in body posture.
Furthermore, our study included a diverse sample population that consisted of males and older individuals. It is well established that HRV values can differ based on gender and age, with males and older individuals often exhibiting lower HRV compared to their male and younger counterparts. Therefore, the inclusion of these subgroups in our study may have contributed to the observed lower HRV values.
In light of these factors, we respectfully disagree with the assertion that our study suffers from methodological flaws that were not adequately addressed. We believe that our explanations provide a valid context for the observed HRV values in our study, given the specific circumstances and demographics of our participants.
However, we are open to further suggestions and guidance from you regarding any specific concerns you have regarding our methodology or analysis. We remain committed to improving the quality of our research and appreciate your expertise in helping us achieve this goal.